# The Construction and Immunoadjuvant Activities of the Oral Interleukin-17B Expressed by *Lactobacillus plantarum* NC8 Strain in the Infectious Bronchitis Virus Vaccination of Chickens

**DOI:** 10.3390/vaccines8020282

**Published:** 2020-06-06

**Authors:** Shaohua Guo, Junjie Peng, Yongle Xiao, Yanyan Liu, Weiwei Hao, Xin Yang, Hongning Wang, Rong Gao

**Affiliations:** 1College of Life Sciences, Sichuan University, Chengdu 610065, China; gsh1366765@163.com (S.G.); pengjunjie93@163.com (J.P.); xiao_yongle@163.com (Y.X.); liuyanyan20010@163.com (Y.L.); yangxin0822@163.com (X.Y.); whongning@163.com (H.W.); 2Sichuan Huapai Biopharmaceutical Company, Chengdu 610026, China; haowei20045841@163.com

**Keywords:** chicken IL-17B, *Lactobacillus plantarum*, oral adjuvant, IBV vaccine, immunity

## Abstract

Interleukin-17B (IL-17B) is a protective cytokine of the IL-17 family and plays an essential role in the regulation of mucosal inflammation. However, little is known about the role of IL-17B in the control of viral infections. In this study, a recombinant *Lactobacillus plantarum*, designated as NC8-ChIL17B, was constructed to express the chicken *IL-17B* (*ChIL-17B*) gene. The recombinant ChIL17B (rChIL17B) protein was about 14 kDa and was anchored to the surface of NC8 cells. In vitro, it was found that the rChIL17B protein inhibited the proliferation of the infectious bronchitis virus (IBV) through activation of nuclear factor kappa B (NF-κB) and the JAK (Janus kinase)-STAT (signal transducers and activators of transcription) signaling. Moreover, to evaluate the immunoadjuvant activities of NC8-ChIL17B, 40 three-day-old specific pathogen-free (SPF) chickens were divided into four groups. Three groups were orally vaccinated with fresh NC8, NC8-ChIL17B, and phosphate buffered saline (PBS), along with the infectious bronchitis virus vaccine, and the other group was the PBS-negative control. The results of the IBV-specific antibody titer and the concentration of the cytokines IL-2, IL-4, IL-6, and interferon gamma (IFN-γ) in sera, as well as the concentration of secretory immunoglobulin A (sIgA) in the tracheal and small intestinal mucosa, the number of cluster of differentiation 4 positive (CD4^+^) and cluster of differentiation 8 positive (CD8^+^) T cells in the blood, and the expression of immune-related genes all indicated that NC8-ChIL17B efficiently enhanced the humoral and cellular immune responses to IBV vaccine. Moreover, the viral loads in the NC8-ChIL17B- and IBV-vaccinated group were significantly lower than in the control groups, suggesting a significant promotion of the immunoprotection of IBV vaccination against the virulent IBV strain. Therefore, ChIL-17B is a promising, effective adjuvant candidate for chicken virus vaccines.

## 1. Introduction

Infectious bronchitis virus (IBV), an enveloped virus with single-stranded RNA, is the major causative pathogen of avian infectious bronchitis (IB), which is a commonly occurring, highly contagious avian disease that causes huge losses in the poultry industry worldwide due to prevalent mixed infection with other pathogens [1]. Vaccination is the most effective measure to control IBV infection. However, the traditional IBV vaccines are not effective enough due to the large number of IBV serotypes. Additionally, new types of vaccines, such as the multi-epitope-based vaccine, cannot provide adequate protection, producing a lower antibody titer and a poor cellular immune response. An adjuvant can greatly change the immune responses and protection associated with vaccines, and can reduce the immunological stress of vaccines and extend the duration of effective immunity [2,3]. At present, the most approved adjuvants for veterinary vaccines are aluminum and oil-emulsion adjuvants. Aluminum adjuvants are barely able to induce cell-mediated immunity, especially cytotoxic T-cell responses, and may promote immunoglobulin E (IgE)—mediated anaphylaxis [4]. Oil-emulsion adjuvants can cause inflammation, ulcers, fever, and sensitivity at the injection site [5]. These adverse reactions affect the appearance and quality of meat and cause discomfort to animals, which is not conducive to animal welfare. Therefore, there is an urgent need to develop new, safe, and efficient adjuvants.

Cytokines play a crucial role in controlling the immune response. Previous studies have indicated that cytokines such as chicken interferon gamma (IFN-γ) [6], interferon α (IFN-α) [7], interleukin-1β (IL-1β) [8,9], interleukin-2 (IL-2) [10], interleukin-18 (IL-18) [11,12,13,14], interleukin-6 (IL-6) [15], interleukin-4 (IL-4) [16], and interleukin-15 (IL-15) could be promising adjuvants, due to their safety and high efficiency [17]. The co-administration of antigens with cytokines is a promising strategy to reduce the side-effects of adjuvants, as well as to enhance humoral and cell-mediated immune responses to promote vaccine efficacy.

IL-17B is a member of the IL-17 family and is an anti-inflammatory cytokine. In mammals, IL-17B is highly expressed in chondrocytes, intestinal epithelial cells, neurons, and breast cancer cells [18,19], and IL-17 receptor B (IL-17RB) is expressed by mucosal epithelial cells [20,21,22]. Although the role of IL-17B in pathogen infection is still not clear, it has been confirmed that IL-17B is protective and plays an important role in the regulation of mucosal inflammation [23]. IL-17B antagonizes IL-17E functions in colitis, extracellular bacterial infections, and allergic asthma [18]. Recently, the full length of the chicken IL-17B (ChIL-17B) coding sequence in small intestinal epithelial cells was reported and characterized [24]. Additionally, ChIL-17B produced by *Escherichia coli* can induce pro-inflammatory cytokines through activation of the nuclear factor kappa B (NF-κB) signaling pathway in vitro. Taken together, this highlights ChIL-17B as an important cytokine for defense against pathogens in chickens. However, little is known about the function of ChIL-17B in the case of viral infections, such as IBV infection, or about its immunoregulation in IBV-vaccinated chickens.

Probiotic *Lactobacillus*, which can tolerate gastric acid and bile salts and stay in the intestines for several days, are considered ideal carriers for mucosal administration. *L. plantarum* NC8 strain is a nonpathogenic, nonsporulating, nonplasmid bacterium that has been reported to be an effective delivery system for HA2 of the H9N2 influenza virus [25], the H1N1 M2e antigen [26], and the HN of Newcastle disease virus (NDV) [27] in chickens. Moreover, oral administration is kinder to chickens and saves time and labor under intensive conditions. Therefore, in order to develop a safe and efficient adjuvant for the chicken IBV vaccine, we inserted chicken IL-17B into *Lactobacillus plantarum* NC8 sequence and evaluated the effect of recombinant NC8-ChIL17B against IBV infection in vitro, as well as its immunoadjuvant activities on the IBV vaccine in specific pathogen-free (SPF) chickens.

## 2. Materials and Methods

### 2.1. Bacteria and Plasmids

The *L. plantarum* NC8 strain, provided by Professor Chunfeng Wang from Jilin Agricultural University, was used as the host. The NC8 strain is a plasmid-free strain isolated from silage and can be grown aerobically in de Man, Rogosa and Sharpe (MRS) medium (Solarbio Science & Technology Co., Ltd., Beijing, China) at 30 °C without shaking [28]. The plasmid pMG36e [29], an *E. coli*–*Lactobacillus* shuttle plasmid (BioVector NTCC Inc., Beijing, China), was duplicated in *E. coli* DH5α strain and used to carry the exogenous gene into the *L. plantarum* NC8 strain and to express the exogenous protein without an inducer.

### 2.2. Construction of Recombinant L. Plantarum NC8-ChIL17B

The sequence for chicken *IL-17B* was obtained from GenBank (Accession No. XM_015293704.1). The sequence was codon-optimized to exploit the natural codon preference of *L. plantarum* and to enhance its protein expression. Furthermore, the signal peptide sequence SPUsp45 from *Lactococcus lactis* MG1363 strain (GenBank Accession No.AM406671, site 2462440-2463825) was linked to the 5′ end of *ChIL-17B* by two repeat linker gene fragments (GGTTCTGGTGGTTCTGGTTCTGGTGGTTCT), in order to promote the ChIL-17B protein’s secretion from the cell wall and to make the ChIL-17B function naturally. Furthermore, a 6×His-Tag gene (CACCACCACCACCACCAC) was added to the 3′ end in front of the stop codon for easy detection of the expression of the target protein by the anti-His-Tag antibody. The new gene was named *rChIL-17B* (Figure 1a) and its size was 645 base pairs. The *rChIL-17B* gene was synthesized by the Tsingke Biotechnology Company (Tsingke Biotechnology Company, Chengdu, China). The *rChIL-17B* gene was ligated to multiple cloning sites of plasmid pMG36e as *pMG36e-ChIL17B* (Figure 1b) using the nonligase-dependent ClonExpress^®^ II One Step Cloning Kit (Vazyme Biotech, Nanjing, China), following the manufacturer’s instructions. The ligation product was transformed into *E. coli* DH5α competent cells (Tiangen biotech Co., Ltd., Beijing, China) and cultured on a SOC agar plate (Beijing Solarbio Science & Technology Co., Ltd.) with 200 μg/mL erythromycin (erm). The positive pMG36e-ChIL17B clones were extracted and sequenced by the Tsingke Biotechnology Company (Tsingke Biotechnology Company, Chengdu, China). The recombinant pMG36e-ChIL17B and the wide-type plasmid pMG36e were then transformed into the electrocompetent NC8 strain by Gene Pulser Xcell™ (Bio-Rad, Berkeley, CA, USA) at 10 kV/cm, 25 μF, 200 Ω, with the resulting cells named NC8-ChIL17B and NC8-P, respectively. The colonies able to grow on MRS agar were supplemented with 10 μg/mL erm and incubated at 37 °C overnight. The presence of pMG36e-ChIL17B in the NC8 cells was identified by PCR and agarose gel electrophoresis. The identified primers from the plasmid pMG36e were as follows: Forward primer: Pc-F TACTTTGGATTTTTGTGAGCT; reverse primer: Pc-R TGTCGCTAGTACCGGTTGT.

### 2.3. Analysis of rChIL-17B Expression by Western Blot and ELISA

The preliminary results of the Western blot assay, the whole-cell ELISA assay [30], and the growth curve measurements indicated that rChIL-17B was anchored to the surface of the NC8 cells, and that the best harvest time for the NC8-ChIL17B strain culture was 24 h, when the optical density (OD) of the NC8-ChIL17B culture was at its maximum of about 1.42 under 630 nm wavelength (Appendix A). To detect the concentration of rChIL-7B, 2 mL fresh NC8-ChIL17B and NC8 strain within wild type plasmid pMG36e (NC8-P) carrier control culture containing 3.4 × 10^9^ colony-forming unit (CFU) of each was centrifuged at 4000× *g* for 5 min to collect the pellets, which were then washed twice with sterile PBS. The pellet was resuspended in PBS with 1 mg/mL lysozyme. After freezing and thawing twice at −80 °C, the bacterial fluid was subjected to sonic disruption using a high-performance ultrasonic sample-processing system Covaris S220 (Covaris, Inc., Woburn, Massachusetts, USA). The cell-free extract was then collected by centrifugation and diluted in 200 μL cold PBS, before being semiquantified by Nanodrop 2000 (Thermo Fisher Scientific, Waltham, MA USA) and the His-Tag ELISA Detection Kit (GenScript Biotech, Nanjing, China) following the manufacturer’s instruction. The detection range of this ELISA kit was 1–729 ng/mL. Next, 10 μL of the extract fluid was added to 10 μL 2× loading buffer and then separated by 12% sodium dodecyl sulfate–polyacrylamide gel electrophoresis (SDS-PAGE), followed by Western blot assay. The mouse anti-His-Tag monoclonal antibody (Abcam Ltd., Shanghai, China) was used as the primary antibody, and the horseradish peroxidase (HRP)-labeled goat anti-mouse immunoglobulin G with heavy chain and light chain [IgG (H + L)] (Proteintech Group, Inc., Wuhan, China) was used as the secondary antibody. The beyoECL plus substrates (Beyotime Biotechnology Ltd., Shanghai, China) were then used to observe the immunoreactive ban under the Bio-Bad ChemiDoc Touch system (Bio-Rad, Berkeley, CA, USA).

### 2.4. Analysis of the rChIL-17B Activity on the Proliferation of IBV in Vitro

The HD11 cell line, a chicken macrophage cell line, was cultured in complete dulbecco’s modified eagle medium (DMEM) (HyClone Inc., Logan, Utah, USA) containing 10% heat-inactivated fetal bovine serum (Gibco, Grand Island, NY, USA) in a humidified 5% CO_2_ incubator at 41 °C. Approximately 1.0 × 10^6^ cells were incubated in a six-well plate (Corning Inc., Corning, NY, USA) for more than 16 h, until they covered 70% of the plate wells. Next, 50 mL of NC8-ChIL17B fresh culture and 50 mL of NC8-P fresh culture were collected by centrifugation and resuspended in 5 mL cold PBS solution, before being disrupted using the high-performance ultrasonic sample-processing system Covaris S220 (Covaris, Inc., Woburn, Massachusetts, USA). The bacterial protein mixture was filtered by a 0.45-nm sterile filter to remove bacteria, and then the concentration of the rNC8-ChIL17B filtrate was detected by using Nanodrop 2000. The mixed filtrates were diluted to 200 ng/mL with 220 picogram (pg) rChIL-17B, which was the optimal nontoxic dose for IBV-infected HD11 cells, verified using the MTT Cell Proliferation and Cytotoxicity Assay Kit (Beyotime Biotechnology Ltd., Shanghai, China) (Appendix A). The cells were then stimulated with the NC8-ChIL17B dilution, followed with 2 MOI (multiple of infection) IBV Beaudette (IBV B) 2 h later. The cell wells with PBS and IBV only were used as the negative and positive controls, respectively. Additionally, the 200 ng/mL NC8-P dilution was used as the host control. Each treatment was repeated in six wells. When the positive IBV control group showed an obvious cytopathic effect (CPE), the cell cultures were collected to determine the median tissue culture infective dose (TCID_50_) using the Reed–Muench method.

### 2.5. Animal Experimental Protocol

Specific-pathogen-free (SPF) White Leghorn chicken embryos were purchased from Beijing Boehringer Ingelheim Vital Biotechnology Co., Ltd. (Beijing, China) and hatched by Sichuan HuaPai Bio-Pharmaceutical Co., Ltd., (Sichuan, China). Forty three-day old chickens were randomly divided into four groups, namely the treatment group NC8-IL17B + IBV and the control groups NC8-P + IBV, IBV, and PBS. Chickens in the NC8-IL17B + IBV, NC8-P + IBV, and IBV groups were orally inoculated with the fresh NC8-ChIL17B strain (1.0 × 10^9^ CFU), NC8-P (1.0 × 10^9^ CFU), and 200 μL PBS, respectively. At the same time, the three groups were orally vaccinated with attenuated IBV H120 (200 μL, 1.0 × 10^8^ EID_50_/mL), provided by Sichuan HuaPai Bio-Pharmaceutical Co., Ltd., Chengdu, China. The PBS control group was orally incubated with 200 μL PBS buffer only. On the 14th day post-primary vaccination (dpv), boost vaccinations were performed with the same protocol as the primary vaccination. On the 0, 7th, 14th, 21st, and 28th dpv, the weight of every chicken was recorded to evaluate growth performance.

On the 28th dpv, five chickens were randomly euthanized to collect the tracheas, thymus, cecal tonsils, and small intestines to analyze the expression of immune-related genes and the concentration of mucosal secretory immunoglobulin A (sIgA). Additionally, five chickens in each group, namely NC8-IL17B + IBV, NC8-P + IBV, and PBS + IBV, were intranasally challenged with 1 mL 1.0 × 10^5.8^ EID_50_ (50% egg infective dose) IBV M41, provided by Chengdu TECH-BANK Biological Products Co., Ltd. (Chengdu, China). The PBS group was incubated with SPF chicken embryo allantoic fluid. The challenged chickens were observed daily for 10 days. All of the chickens were then euthanized to obtain tissue samples (i.e., lungs, livers, spleens, kidneys, bursae, and cecal tonsils) to quantify the replication of virulent IBV M41.

The animal experiments in this study were approved by the Animal Ethics Committee (ACE) of the Animal Experiment Center of Sichuan University (license: SYXK-Chuan-2018-185). All experimental procedures and animal welfare standards strictly followed the animal management guidelines of Sichuan University.

### 2.6. Evaluation of the Immunoadjuvant Effect of NC8-ChIL17B In Vivo

#### 2.6.1. Measures of IBV-Specific Antibodies and the Concentration of Cytokines in the Serum by ELISA

On the 0, 7th, 14th, 21st, and 28th dpv, a 200-μL blood sample was collected from each chicken to detect any changes of IBV-specific antibody titer and the concentrations of cytokines IL-2, IL-4, IL-6, and IFN-γ in the serum. The Infectious Bronchitis Virus Antibody Test Kit (IDEXX, Westbrook, ME, USA) was employed to determine the IBV-specific antibodies in the serum according to the manufacturer’s instructions. The concentrations of IL-2, IL-4, IL-6, and IFN-γ in the serum were individually monitored using the corresponding sandwich ELISA kit (SinoBestBio, Shanghai, China).

#### 2.6.2. Measures of sIgA from the Tracheas and Small Intestines by ELISA

On the 28th dpv, 8-cm-long fresh tracheas and 10-cm samples of the small intestines (including the Peyers’ patches) were repeatedly washed with 500 μL cold PBS containing protease inhibitor to obtain lavage fluid, and were then centrifuged at 10,000× *g* for 20 min at 4 °C to remove precipitated impurities before being stored at −20 °C. The concentrations of total sIgA in the lavage fluid were individually monitored using the direct sandwich ELISA kit (SinoBestBio, Shanghai, China) according to the manufacturer’s instructions. For detection of the titer of IBV-specific sIgA in the lavage fluid, IBV-coated 96-well ELISA plates were used to bind the IBV-specific sIgA with 100-μL lavage for each well and three-well repeats for each sample, which were then incubated at 37 °C for 1 h. After washing five times, 100-μL goat anti-chicken IgA H&L (HRP) (Abcam Ltd., Shanghai, China) at a 1:2000 dilution was added to each well and incubated at 37 °C for 1 h, followed by washing. Finally, 10 μL TMB (3,3′,5,5′-Tetramethylbenzidine dihydrochloride hydrate) Chromogen Solution (Beyotime Biotechnology Ltd., Shanghai, China) was added to each well, which were incubated at room temperature for 10 min without light, and then the plates were read at the wavelength of 630 nm.

#### 2.6.3. Analysis of CD4^+^ and CD8^+^ T Cells in the Peripheral Blood by Flow Cytometry (FCM)

The peripheral blood lymphocytes (PBLCs) were isolated from the sodium citrate anticoagulant by using the chicken peripheral blood lymphocyte isolation kit (TBD Sciences Inc., Tianjin, China) according to the manufacturer’s instructions. Approximately 1.0 × 10^6^ PBLCs mixed with 100 μL normal saline were then incubated with 5 μL PE (phycoerythrin)-conjugated CD4 monoclonal antibody (CT-4) and 10 μL APC (allophycocyanin) -labeled CD8 monoclonal antibody (Invitrogen, Carlsbad, CA, USA) for 30 min. Following this, the cells were analyzed using the FACScan flow cytometer (Becton, Dickinson and Company, Franklin Lake, NJ, USA).

#### 2.6.4. Analysis of Immune-Related Genes by Quantitative Real-Time Polymerase Chain Reaction (qRT-PCR)

To analyze immune-related gene expression, 0.1 g of each tissue sample (i.e., thymus and cecal tonsils) was added to 500 μL cold PBS buffer and homogenized by a high-throughput tissue grinder (DHS life science, Beijing, China). Next, 200 μL of tissue homogenate was added to 800 μL Trizol (Invitrogen, Carlsbad, CA, USA) to extract RNA, following the appropriate instructions. The total RNA was reverse-transcribed at 42 °C for 15 min using TransGen, TransScript All-in-One First-Strand cDNA Synthesis SuperMixforqPCR (One-Step gDNA Removal) (Transgen, Beijing, China), as described in the manufacturer’s guide. Nine pairs of primers for the PCR of immunity-related genes were designed and synthesized based on the relevant gene sequences in GenBank (Table 1 and Appendix A). Real-time PCR was performed using Bio-Rad CFX connect (Bio-Rad, Berkeley, CA, USA) in a total reaction volume of 20 μL, consisting of 5 μL of 20-fold diluted complementary DNA (cDNA), 10 μL of AceQ^®^ qPCR SYBR Green Master Mix (Vazyme Biotech Co., Ltd., Nangjing, China), 0.5 μL of each forward and reverse primer (10 μmol/L), and 4 μL of sterile ddH_2_O. The PCR reaction protocol was carried out with an initial denaturation for 5 min at 95 °C, followed by 40 cycles of 10 sec at 95 °C and 30 sec at 55 °C, with a melt curve at the end of all cycles. The samples from the PBS group were the negative control included in each run. The β-actin gene was used as a reference gene. The messenger RNA (mRNA) fold change of the immune-related genes were calculated via the geometric means method 2^− ΔΔCt^ [31]. ΔCt = Ct (target gene) − Ct (reference gene); ΔΔCt = ΔCt (sample) − ΔCt (control).

#### 2.6.5. Measurement of IBV M41 Copies in Challenged Chickens by Absolute qRT-PCR

In order to evaluate the viral RNA levels in the tissues (i.e., tracheas, lungs, spleens, bursae, kidneys, and livers), 0.1 g of each tissue sample was used for RNA extraction, using the method described in Section 2.6.4. After reverse transcription (total volume of 10 µL), 1 µL of the reverse transcription product (i.e., one-tenth of the total RNA) was used as a template in absolute quantification real-time PCR (qPCR) to determine the IBV copies. Primers were designed using Primer Premier version 6 (Premier Inc., Palo Alto, CA, USA) according to the genome sequences of IBV M41 (GenBank accession No. AY851295.1) and IBV H120 (GenBank accession No. FJ888351.1). The primer sequences were as follows: P_IBV_-F 5′-TCTGAGAAATCAGTTGAGGGT-3′ and P_IBV_-R 5′-ACTCATCAACCTCTTCTGCTG-3′. Six repeats were carried out for each sample. Real-time PCR was performed using Bio-Rad CFX connect (Bio-Rad, Carlsbad, CA, USA) in a total reaction volume of 20 μL, consisting of 1 μL of cDNA, 10 μL of AceQ^®^ qPCR SYBR Green Master Mix (Vazyme Biotech Co., Ltd., Nangjing, China), 0.5 μL of each forward and reverse primers (10 μmol/L), and 8 μL of sterile ddH_2_O. The PCR reaction protocol was carried out with an initial denaturation for 5 min at 95 °C, followed by 40 cycles of 10 sec at 95 °C and 30 sec at 55 °C, and a melt curve at the end of all cycles. The DNA fragment from the IBV M41 genome (Sites 2870–3420) was inserted into the plasmid pUC57 plasmid to draw the standard curve to quantify the loads of IBV M41 RNA in the challenged chicken tissues.

### 2.7. Statistical Analysis

GraphPad Prism 6 (GraphPad Software Inc., La Jolla, CA, USA) was used for data analysis and graphical presentations.

## 3. Results

### 3.1. Construction of the NC8-ChIL17B Strain

As shown in Figure 2a, the *rChIL-17B* gene was successfully transformed into the NC8 strain. At the optimal harvest time, the mass of rChIL-17B was 3.52 ng in about 1.2 × 10^9^ CFU. However, the concentration of rChIL-17B in the NC8-P culture was under the detection limit. The Western blot results showed that a protein band of approximately 14 kDa (Figure 2b), the predicted size of the rChIL-17B protein, was detected using the mouse anti-His-Tag monoclonal antibody. Additionally, the NC8-P had no specific band. Taken together, this indicated that rChIL-17B was successfully expressed by NC8-ChIL17B.

### 3.2. Analysis of the rChIL17B Activity on the Proliferation of the IBV Beaudette Strain in HD11 Cells

Table 2 shows the proliferative titers of the IBV B strain on the HD11 cells after being treated with 55 pg/mL rChIL17B solution for 48 h when the IBV-infected cells’ control showed obvious cytopathic effect (CPE). The IBV titer in the rChIL17B-treated cells were 10^4.65 ± 0.11^ TCID_50_/mL, which was significantly lower than in the IBV control cells (*p* < 0.01) and the NC8-P + IBV control cells (*p* < 0.01).

### 3.3. Weight Change in the Experimental Chickens

Table 3 shows the mean weight gains of each chicken in the different treatment groups. By the end of the 28th dpv, the chickens in the NC8-ChIL17B + IBV group had gained more weight than the chickens of the NC8-P + IBV group, as determined using an unpaired *t*-test (*p* < 0.01), although the initial weights of the two groups had no significant difference. The initial weight and the end weight of the PBS and IBV groups were not significantly different.

### 3.4. Evaluation of the Immunoadjuvant Activities of NC8-ChIL17B in Chickens

#### 3.4.1. Analysis of the IBV-Specific Antibodies in the Serum

On the 7th, 14th, 21st, and 28th dpv, the IBV-specific antibodies in the serum were evaluated by indirect ELISA. Starting from the 7th dpv, the sample vs positive control (S/P) values of the IBV-specific antibodies in the IBV, NC8-P + IBV, and NC8-ChIL17B + IBV groups were >0.2, which means that the IBV vaccine induced effective antibodies. Additionally, throughout the 28-day vaccination period, the IBV-specific antibody titers in the NC8-ChIL17B + IBV group were significantly higher than those in the IBV and NC8-P + IBV groups: *p* < 0.05 on the 7th and 28th dpv, and *p* < 0.01 on the 14th and 21st dpv (Figure 3). Furthermore, on the 14th dpv, the IBV-specific antibody level reached the highest titers.

#### 3.4.2. Changes in the Concentrations of Cytokines in the Serum

As shown in Figure 4, at 0 dpv, the concentrations of IL-2, IL-4, IL-6, and IFN-γ showed no differences across the four groups. The concentration of IL-2 (Figure 4a) in the NC8-ChIL17B-IBV group was significantly higher than in the IBV and NC8-P + IBV groups (*p* < 0.05) on the 14th and 28th dpv. Furthermore, the concentration of IL-2 in the IBV and NC8-P + IBV groups was also significantly higher than in the PBS control group: *p* < 0.05 on the 14th dpv, and *p* < 0.01 on the 28th dpv. Moreover, the concentrations of IL-4 (Figure 4b), IL-6 (Figure 4c), and IFN-γ (Figure 4d) in the NC8-ChIL17B group were very significantly higher compared to that from the IBV, NC8-P + IBV, and PBS groups (*p* < 0.01) on the 14th and 28th dpv (*p* < 0.01).

#### 3.4.3. Changes in the CD4^+^ and CD8^+^ T-Cell Counts in the Peripheral Blood

The CD4^+^ and CD8^+^ T-cell counts in the peripheral blood were determined using flow cytometry (FCM). On the 14th and 28th dpv, a very significant upregulation of CD4^+^ T cells (Figure 5a) and CD8^+^ T cells (Figure 5b) was observed in the chickens from the NC8-ChIL17B + IBV group compared to those in the IBV and NC8-P + IBV groups (*p* < 0.001). Additionally, the CD4^+^ and CD8^+^ T cells in the NC8-P + IBV group were significantly increased in comparison to the IBV group. The CD4^+^/CD8^+^ ratio in all three groups was >2.0, and on the 28th dpv the ratio in the NC8-ChIL17B + IBV group was >3.0, which was significantly higher than that in the IBV group (*p* < 0.05) but not significantly different in comparison to the NC8-P + IBV group.

#### 3.4.4. Changes in Mucosal sIgA in the Tracheas and Small Intestines

The concentrations (μg/mL) of total sIgA in the trachea and intestine lavage solutions were detected using sandwich ELISA. Additionally, the IBV-specific sIgA titers were detected by indirect ELISA. In the tracheas, both the total sIgA and the IBV-specific sIgA in the NC8-ChIL17B + IBV group were significantly higher than in the IBV and NC8-P + IBV control groups (Figure 6a,c), *p* < 0.05. In the intestines, both the total sIgA and the IBV-specific sIgA of the NC8-ChIL17B + IBV group were significantly higher than in the control groups: *p* < 0.01 in the tracheas and *p* < 0.05 in the intestines (Figure 6b,d). Furthermore, in the IBV and NC8-P + IBV groups, the concentrations of total sIgA and IBV-specific sIgA did not significantly differ in either the tracheas or the intestines. Moreover, the concentration of total sIgA was higher in the tracheas than that in the intestines.

#### 3.4.5. Changes of the Expression of Immune-Related Genes

The mRNA levels of the IL-1β, Toll-like receptor (TLR-3), B-cell CLL/lymphoma 6 (BCL-6), IL-6, transforming growth factor-β 4 (TGF-β4), TLR-7, cluster of differentiation 127 (CD127), and IL-22 genes in the thymus and cecal tonsils from the experimental chickens were analyzed by qRT-PCR. As shown in Figure 7, the expression of the cytokine genes, IL-1β and IL-6, and the BCL-6 gene, which participates in regulating cell and/or humoral immunity, was significantly upregulated in the NC8-ChIL17B + IBV group compared to the PBS, IBV, and NC8-P + IBV control groups, either *p* < 0.05 or *p* < 0.01. The expression of IL-22 is important for mucosal barriers. In the thymus, the expression IL-22 of the NC8-ChIL17B + IBV, NC8-P + IBV, and IBV groups was significantly upregulated compared to the PBS control group (*p* < 0.05), but there were no significant differences between the three groups. In the cecal tonsils, the expression of IL-22 in the NC8-ChIL17B + IBV group was significantly upregulated compared to the control groups (*p* < 0.05). The expression levels of both TLR-3 and TLR-7 in the thymus and the cecal tonsils, which are involved in innate immunity, were significantly upregulated in the NC8-ChIL17B + IBV group compared to the control groups, either *p* < 0.05 or *p* < 0.01. The expression of the TGF-β4 gene, one of the cytokines that participates in maintaining immune homeostasis, was also significantly upregulated compared to the three control groups: *p* < 0.05 in the thymus and *p* < 0.01 in the cecal tonsils. The expression of CD127, the marker of memory CD8^+^ T cells, was significantly upregulated compared to the control groups: *p* < 0.01 in the thymus and *p* < 0.05 in the cecal tonsils, and the gene expression fold-change in the thymus was higher than in the cecal tonsils.

#### 3.4.6. Evaluation of the Protection Against IBV M41 Challenge

In this study, chickens were challenged with the virulent IBV M41 strain and kept for 10 days to observe their clinical symptoms. Under virulent IBV strain challenge, the chickens in the PBS group presented apparent cough and asthma in the morning. However, there were no apparent symptoms observed in the chickens of the IBV, NC8-P + IBV, or NC8-ChIL17B + IBV groups. To evaluate the resistance effect of the IBV H120 vaccine to the virulent M41 strain, the IBV M41 genome copies in the tissues (i.e., tracheas, lungs, spleens, kidneys, bursae, and livers) on the 10th day post-challenge (dpc) were analyzed by absolute qPCR. As shown in Figure 8, the mean IBV M41 copies in the IBV-immunized groups, namely IBV, NC8-P + IBV, and NC8-ChIL17B + IBV, were very significantly reduced compared to those of the PBS group (*p* < 0.01). The mean IBV M41 copies in the tracheas, lungs, spleens, kidneys, and bursae from the NC8-ChIL17B + IBV group were very significantly or significantly lower compared to the NC8-P + IBV control group: *p* < 0.01 in the lungs and *p* < 0.05 in the tracheas, spleens, kidneys, and bursae.

## 4. Discussion

In this study, the recombinant *L. plantarum* NC8-ChIL17B strain, which can secrete the bioactive IL-17B, was developed as an oral immunoadjuvant for IBV vaccine. The *L. plantarum* carrier was favorable for solving the problem of the short half-life of cytokines. *L. plantarum* can effectively stimulate local and systemic immunity in animals [32], which was also observed in the NC8-P + IBV group in this study. Here, the rChIL-17B protein with SPUsp45 signal peptide sequence from *Lactococcus lactis* was anchored to the NC8 cell surface instead of being secreted into the medium. This surface-display expression method can enrich foreign proteins and enhance their bioactivity [33,34,35].

In this study, it was found that rChIL-17B obviously inhibited the proliferation of IBV in HD11 cells. To investigate the potential mechanism of rChIL-17B and its effects on IBV proliferation, as well as its immunoregulation effect, the production of immune-related genes in rChIL-17B-treated HD11 cells and DF-1 cells (a chicken fibroblast cell line) was detected using qRT-PCR (Appendix A). The upregulation of genes in the NF-κB signaling pathway (NF-κB, myd88, and TAK1 SOCS1), the the JAK (Janus kinase)-STAT (signal transducers and activators of transcription) (JAK/STAT) signaling pathway (JAK2, TYK2, STAT1, and STAT3), and their downstream cytokines (IL-1β, IFN-α, TGF-4β, IL-6, IL-4, IFN-γ, IL-12P40, IL-10, chemokines 3 (CCL3), and CCL20) indicated that rChIL-17B regulated the immune response through the JAK/STAT and NF-κB signaling pathways to activate downstream Th1 and Th2 cytokine expression. The JAK/STAT signaling pathway provides a direct mechanism for cytokines to regulate gene expression in immune regulation [36]. The rChIL-17B expressed by *E. coli* induced pro-inflammatory cytokines through activation of the NF-κB signaling pathway, as previously reported [26]. Moreover, we found that the mRNA levels of the IFN-γ and Bcl-2 genes in IBV-infected HD11 cells were upregulated, and the expression of the FasL gene was downregulated (Appendix A). Bcl-2 is an important apoptosis suppression gene. The FasL gene participates in the induction of cell apoptosis. In particular, IBV B was able to induce apoptosis of HD11 cells in a previous study [37]. Furthermore, in mammals, the IL-17B/IL-17 BR complex activates NF-κB and upregulates its anti-apoptotic pathway [38]. Taken together, this indicates that IL-17B can inhibit IBV proliferation through upregulating immune-related gene expression and inhibiting the apoptotic pathway.

During the animal experimental period, the chickens in the NC8-ChIL17B + IBV group displayed obvious improvements in growth compared to the NC8-P+IBV group. Although it is common sense that the probiotic effect of *L. plantarum* promotes growth performance by changing the microbial balance of the intestinal flora, the expression of cytokines such as IL-2, IL-4, IL-6 IFN-γ, and IL-22, which participate in the cytokine–endocrine hormone interaction network, affect the growth of epithelial cells in the digestive tract, improving digestion and metabolism [39,40]. Similar results were observed in our previous study, in that the fusion cytokine IL-4/6 promoted the growth performance of piglets [41]. The results indicate that rChIL17B has an effect on promoting weight gain.

IBV-specific antibodies are the major indexes of humoral immune responses used in evaluation of IBV vaccines. During the 28-day period, NC8-ChIL17B induced a significant increase in the IBV-specific antibodies compared to the other three control groups. The sIgA level is particularly important for evaluating oral vaccines. The increases of total sIgA and of IBV-specific IgA in the tracheas and intestines showed that the oral IBV vaccine induced mucosal antibodies, which are helpful for resisting virulent IBV strain infection. In clinical vaccination contexts, oral vaccine (attenuated) is used as the first vaccination, and the intramuscular vaccine (inactive) is chosen to boost immunity. However, in this study, it was obvious that NC8-ChIL17B significantly enhanced sIgA production in the mucosal system. IL-17 receptor B (IL-17RB) is expressed by mucosal epithelial cells [20,21,22], which can combine with rChIL-17B to activate the immune system. Moreover, the RNA levels of IL-22 in the thymus and the cecal tonsils of NC8-ChIL17B + IBV were significantly upregulated in this experiment. IL-22 is considered a promising target for tissue-protective therapy, especially in acute disease of epithelial tissues, such as ulcerative colitis or hepatitis [42]. The results indicated that rChIL17B can not only enhance humoral immunity, but also mucosal immunity.

The concentrations of IL-2, IL-4, IL-6, and IFN-γ in the sera of NC8-ChIL17B-treated chickens were significantly higher than in the control groups. IL-2 and IFN-γ are produced by Th1 cells to induce macrophage activation, delayed-type hypersensitivity, and the production of IgG [43,44,45]. IFN-γ is a Type II interferon, which is produced by activated T cells, natural killer cells (NK cells). IFN-γ has antiviral and immunomodulatory characteristics, and is a marker cytokine of Type I helper T cells (Th1 cells). IL-4 and IL-6 are produced by Th2 cells to induce B-cell differentiation to secrete IgM, IgG, and IgA, monocyte proliferation, and neutrophil recruitment to the sites of inflammation [46]. The expression of IL-2 and IL-6 in the thymus and cecal tonsils of the NC8-ChIL17B + IBV group was also significantly upregulated compared to that of the control groups in this experiment. These results indicated that NC8-ChIL17B enhanced the antibody-mediated Th2 response and the cell-mediated Th1 response to the IBV vaccine [43].

Additionally, the CD4^+^ and CD8^+^ T-cell levels in the peripheral blood were significantly raised in the NC8-ChIL17B + IBV group compared to the control groups on the 14th and 28th dpv. CD4^+^ T cells are activated and differentiated into different effector subtypes by the interaction of the T-cell receptor (TCR) and the major histocompatibility complex (MHC) Class II molecular complex. The immune response is mediated by the secretion of specific cytokines. CD8^+^ T cells can recognize antigen–MHC Class I molecular complexes, can differentiate into cytotoxic T cells (CTLs) after activation, and can specifically kill virus-infected target cells by expressing granzymes and perforins, as well as by secreting cytokines such as IFN-γ and tumor necrosis factor (TNF). Memory CD8^+^ T cells are a major component of immunity against intracellular pathogens, like viruses, and CD4^+^ T cells are necessary for the development of memory CD8^+^ T cells during immunity and pathogen infection.

Moreover, the mRNA levels of IL-1β, TLR-3, TLR-7, TGF-β4, BCL6, and CD127 in the thymus and cecal tonsils of the NC8-ChIL17B + IBV group were also significantly upregulated compared to the NC8-P + IBV, IBV, and PBS control groups. IL-1β is a pro-inflammatory factor that promotes antibody production and induces IL-2 production, mainly through T-cell helper cells. TLR-3 and TLR-7 are members of the TLR family. TLR activation promotes both innate inflammatory responses and the induction of adaptive immunity.TLR-3 recognizes the viral replication product dsRNA (a common intermediate of replication among many viruses) [44], and TLR-7 recognizes single-strand (ss) RNA viral infections and activates the related antiviral immune responses. TLR-7 agonists can promote the expansion of memory T cells, B cells, and the long-term protective immune response [45]. TGF-β regulates adaptive immunity and the innate immune system by controlling the generation and effector functions of many immune cell types, including Treg cells, effector T cells, antigen-presenting dendritic cells (DCs), natural killer (NK) cells, macrophages, and neutrophils, thus forming a network of negative immune regulatory inputs [46]. The transcriptional repressor Bcl-6 directs follicular helper T-cell lineage commitment. Follicular helper T (Tfh) cells provide selection signals to germinal center B cells, which are essential for long-lived antibody responses. CD127, one of the two unique α chains in the IL-7 receptor, dictates T cell sensitivity to IL-7. As a surface marker of CD8^+^ T-cell differentiation, CD127 can be expressed on the surface of virus-specific CD8^+^ T cells. A previous study reported that most hepatitis B virus (HBV)-specific CD8^+^ T cells are CD127^+^ subtypes after clearing the virus; thus, CD127 can be used as an early marker of the transformation from effector CD8^+^ T cells to memory CD8^+^ T cells after virus infection [47]. These results indicated that NC8-ChIL17B significantly enhanced the innate immunity, cellular immunity, and immune memory of chickens treated with the IBV vaccine.

A high-quality adjuvant should be able to enhance the resistance of vaccinated chickens against the challenge of virulent strains. Here, the IBV load in the main organs was the key index used to evaluate the protection of the IBV vaccine with NC8-ChIL17B for chickens after challenging them with the IBV M41 strain. IBV M41 is a strongly virulent strain, but it cannot cause obvious clinical symptoms and death in infected chickens. The results showed that the load of IBV M41 was efficiently suppressed in all IBV-vaccinated group compared to the PBS group on the 10th dpc. Additionally, in the NC8-ChIL17B group, the IBV M41 load was the lowest among the four groups. These results indicated that NC8-ChIL17B enhanced the protective effect of the IBV vaccine. It is generally considered that stronger humoral and cellular immunity inhibits the replication of viruses. It has also been confirmed in previous studies that IL-2 can promote the elimination of the Newcastle disease virus (NDV) and can reduce the virus titer of NDV in the blood, spleen, oral secretions, and cloacal excretions [48]. Moreover, the IBV copies in the tracheas were the highest in these tissues, and a similar result was also reported in our previous IBV challenge assay [49].

## 5. Conclusions

We constructed a recombinant *L. plantarum* NC8-ChIL17B-expressing chicken IL-17B (rChIL17B), which is likely a promising adjuvant for IBV vaccines. We first verified that the rChIL17B expressed by NC8-ChIL17B could regulate the immune response through the JAK/STAT and NF-κB signaling pathways and could inhibit the IBV proliferation by inhibiting apoptosis and secreting antiviral cytokines in vitro. The subsequent animal experiment indicated that oral NC8-ChIL17B obviously enhanced the immune response of chickens to the IBV vaccination and protected flocks from IBV M41 challenge, along with an improvement in weight gain. IL-17B has the potential to be developed as a safe and cost-effective adjuvant for IBV vaccines. This is the first study to express avian IL-17B in *L. plantarum* and to employ it as an adjuvant for orally administered IBV vaccine. NC8-ChIL17B could also be developed as an alternative to antibiotics, due to its effect on immune enhancement.

## Figures and Tables

**Figure 1 vaccines-08-00282-f001:**
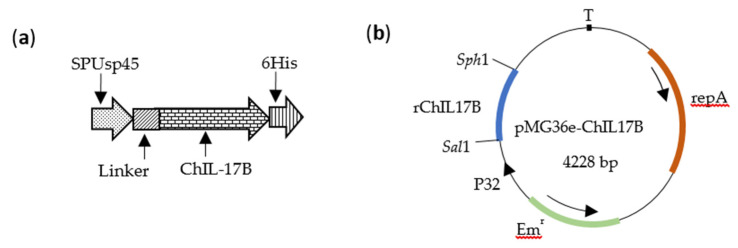
Schematic diagram of the construction of the *rChIL-17B* gene and *pMG36e-ChIL17B*. (**a**) The *rChIL-17B* gene. The signal peptide sequence SPUsp45 and the linker gene fragment (GGTTCTGGTGGTTCTGGTTCTGGTGGTTCT) were used to promote the ChIL-17B protein’s secretion from the cell wall and to make the ChIL-17B function naturally. (**b**) The recombinant plasmid *pMG36e-ChIL17B*. The rChIL17B was inserted into the multiple cloning sites between restriction endonucleases *Sph*1 and *Sal*1.

**Figure 2 vaccines-08-00282-f002:**
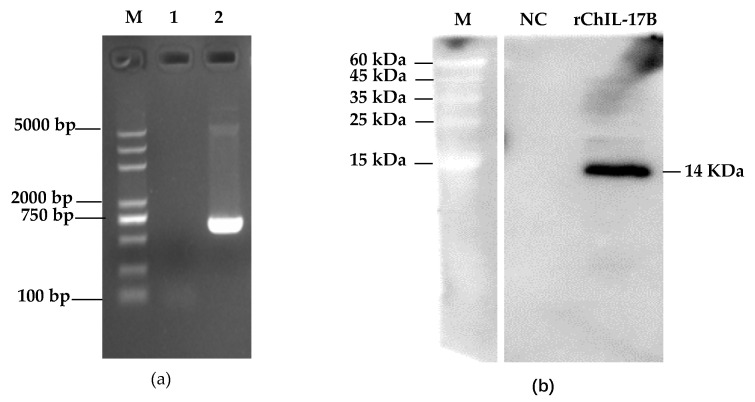
The results of the construction of the NC8-ChIL17B strain. (**a**) Analysis of the *rChIL-17B* gene in NC8 strain by PCR. M: Marker; Lane 1: NC8-P, the wild type plasmid carrier control; Lane 2: NC8-ChIL17B; (**b**) Expression of rChIL-17B analyzed by Western blot. M: protein molecular weight markers; NC: the negative control, strain NC8-P cell protein; rChIL-17B: the recombinant ChIL-17B expressed by *Lactobacillus plantarum* NC8 strain.

**Figure 3 vaccines-08-00282-f003:**
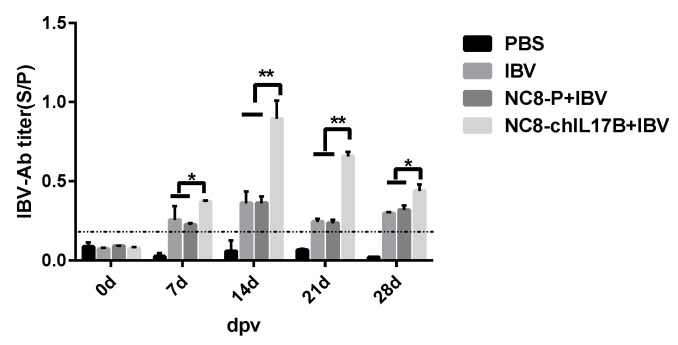
IBV-specific antibodies in the serum. The antibody titers are shown as mean sample vs. positive control (S/P) values + S.D. of each group. The threshold cut-off values of the IBV ELISA were 0.2. S ÷ P = (Sample A (650) − NC A (650)) ÷ (PC A (650) − NC A (650)). Tukey’s multiple comparison test was used for statistical analysis; * *p* < 0.05, ** *p* < 0.01.

**Figure 4 vaccines-08-00282-f004:**
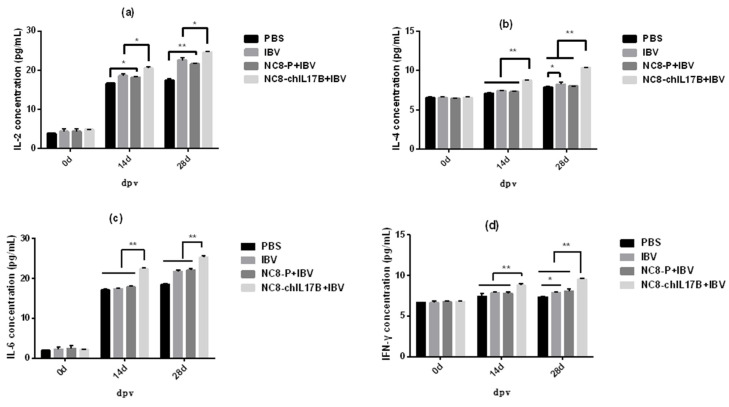
Analysis of the IL-2 (**a**), IL-4 (**b**), IL-6 (**c**), and IFN-γ (**d**) concentrations in the serum on the 0, 14th, and 28th dpv, detected by sandwich ELISA. Tukey’s multiple comparison test was used for statistical analysis. Concentrations are shown as mean values + S.D. of each group; * *p* < 0.05, ** *p* < 0.01.

**Figure 5 vaccines-08-00282-f005:**
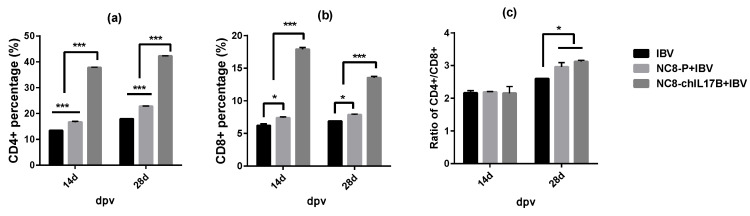
CD4^+^ and CD8^+^ T cells in the blood of the experimental chickens on the 14th and 28th dpv. (**a**) The percentage of CD4^+^ T cells in the peripheral blood of the experimental chickens. (**b**) The percentage of CD8^+^ T cells in the peripheral blood of the experimental chickens. (**c**) The CD4^+^/CD8^+^ ratio. Tukey’s multiple comparison test was used for statistical analysis; * *p* < 0.05, *** *p* < 0.001.

**Figure 6 vaccines-08-00282-f006:**
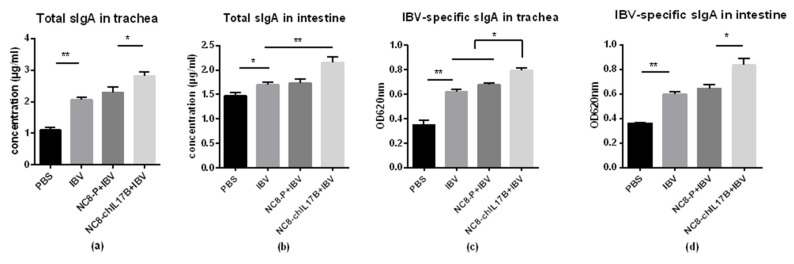
Results of the total sIgA concentrations (**a**,**b**) and the IBV-specific sIgA titers (**c**,**d**) in the tracheas and intestines, detected by ELISA. Concentrations and optical delnsity under 620 nm wavelength (OD_620_) are shown as mean values + S.D. of each group. Tukey’s multiple comparison test was used for statistical analysis; * *p* < 0.05, ** *p* < 0.01.

**Figure 7 vaccines-08-00282-f007:**
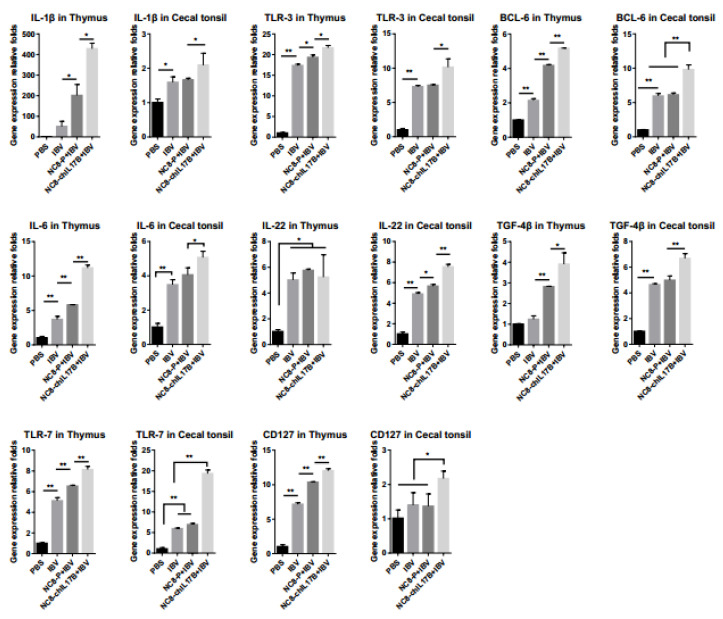
Expression of several immune genes in the thymus and the cecal tonsils. IL-1β: Interleukin-1β; TLR-3: Toll-like receptor 3; BCL-6: B-cell CLL/lymphoma 6; IL-6: Interleukin-6; TGF-β4: transforming growth factor-β 4, TLR-7: Toll-like receptor 7; CD127: cluster of differentiation. Tukey’s multiple comparison test was used for statistical analysis; * *p* < 0.05, ** *p* < 0.01.

**Figure 8 vaccines-08-00282-f008:**
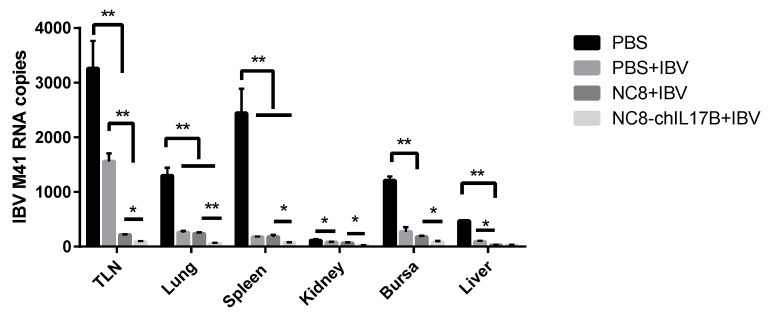
Changes of the IBV M41 RNA copies in the tissues of IBV M41-challenged chickens as determined by absolute quantitative polymerase chain reaction (qPCR). The formula of the standard curve is y = 3.2771x + 37.947, E = 102.0%, R^2^ = 0.9971. A multiple *t*-test was used for statistical analysis; * *p* < 0.05, ** *p* < 0.01.

**Table 1 vaccines-08-00282-t001:** The primers for quantitative real-time polymerase chain reaction (qRT-PCR).

Primers	F/R	Oligonucleotide Sequences (5′–3′)	GenBank Accession No.
β-actin	F	TGCTGTGTTCCCATCTATCG	X00182
β-actin	R	TTGGTGACAATACCGTGTTCA	-
TLR-3	F	GTGCTTGCTAGCTCTCGACT	FJ915480.1
TLR-3	R	GCTTTCTGTGTGCTCCAAGC	-
CD127	F	TGGCATTCAAGCAAAAGCCG	EF116487.1
CD127	R	AATCCTTGCAGGACTTCGCT	-
IL-1β	F	TCGGGTTGGTTGGTGATG	NM_204524
IL-1β	R	TGGGCATCAAGGGCTACA	-
TGF-β4	F	CGTGCCCGTACATCTGGAG	AF459839.1
TGF-β4	R	GAGGGGGTCGAGGGTCTG	-
TLR-7	F	ACGGTGTTGGATCTTGGGAC	NM_001011688.2
TLR-7	R	TGGACTTGCAACTTCGACCA	-
IL-22	F	CAATGCCCATCAAGCCTGCA	AJ617782.1
IL-22	R	CTGTGCCACATCCTCAGCAT	-
BCL-6	F	CTCATCTTCAGACGGCAAAGG	BQ038697.2
BCL-6	R	ATGTCTGTGCAGTGGAGTGTT	-
IL-6	F	CAAGGTGACGGAGGAGGAC	JQ897539
IL-6	R	TGGCGAGGAGGGATTTCT	-

Note: F—forward; R—reverse; TLR—Toll-like receptor; IL—interleukin; CD—cluster of differentiation; TGF—transforming growth factor; BCL—B-cell lymphoma.

**Table 2 vaccines-08-00282-t002:** The infectious bronchitis virus (IBV) proliferative titers in the HD11 cells under different treatments.

Treatment	Lg TCID_50_/mL
PBS	0
IBV	6.58 ± 0.11 ^c^
NC8-P + IBV	6.08 ± 0.11 ^B^
NC8-ChIL17B + IBV	4.65 ± 0.05 ^A^

Tukey’s multiple comparison test was used for statistical analysis. PBS: phosphate buffered saline (PBS) treated cell was taken as negative control; IBV: IBV infected cell was taken as positive control; NC8-P+IBV: NC8 strain within wild type plasmid pMG36e (NC8-P) and IBV treated cell was take as carrier control; NC8-ChIL17B + IBV: NC8-ChIL17B and IBV treated cell group. Different capital letters represent extremely significant differences, *p* < 0.01; different lowercase letters represent significant differences, *p* < 0.05.

**Table 3 vaccines-08-00282-t003:** The chickens’ weight over the 28 days vaccination period.

Group	Initial Weight (g)	End Weight (g)	Net Gain (g)
PBS	62.0 ± 0.76	360.0 ± 5.79	298.0 ± 4.63
IBV	61.0 ± 0.96	350.0 ± 5.92	294.0 ± 5.17
NC8-P + IBV	67.0 ± 0.49	375.0 ± 4.29	309.0 ± 5.52
NC8-ChIL17B + IBV	66.0 ± 0.47	380.0 ± 7.26	314.0 ± 2.98 **

An unpaired *t*-test was used for statistical analysis. PBS: phosphate buffered saline (PBS) treated chickens were taken as negative control; IBV: IBV H120 vaccinated chicken were taken as positive control group; NC8-P+IBV: NC8 strain within wild type plasmid pMG36e (NC8-P) plus IBV H120 vaccine treated chickens were taken as carrier control group; NC8-ChIL17B + IBV: NC8-ChIL17B and IBV H120 vaccine treated chickens group. PBS compared to IBV and NC8-P + IBV compared to NC8-ChIL17B + IBV; *n* = 10 in each group. The weight gains are shown as mean ± SEM; ** *p* < 0.01.

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
