# Peer review of "The Construction and Immunoadjuvant Activities of the Oral Interleukin-17B Expressed by Lactobacillus plantarum NC8 Strain in the Infectious Bronchitis Virus Vaccination of Chickens"

_vaccines, 2020, doi:10.3390/vaccines8020282_

Round 1
Reviewer 1 Report
Construction and immunoadjuvant activities of the oral interleukin-17B expressed by Lactobacillus plantarum NC8 on IBV vaccination of chicken
The authors constructed a recombinant Lactobacillus plantarum NC8 expressing chicken IL-17B gene(NC8-ChIL-17B) and tested biological activity of recombinant ChIL-17B(rChIL-17B) produced by NC8-ChIL-17B. They demonstrated in vitro anti-viral effects of rChIL-17B on IBV-infected HD11 cells and in vivo adjuvant effect of NC8-ChIL-17B on a live IBV vaccine (H120 strain). They compared serum and mucosal antibody, serum cytokines levels, percentages of CD4+, CD8+ T cells, ratio of CD4+/CD8+ T cells, and expression levels of immune-related genes in thymus and cecal tonsils. Also they compared body weight change before and protection efficiency after challenge with M41 strain between PBS, PBS+IBV, NC8+IBV, and NC8-ChIL-17B+IBV groups. Although they found significant difference between NC8+IBV, and NC8-ChIL-17B+IBV groups in vitro and most of in vivo tests they could not demonstrate significant difference between the groups in terms of M41 RNA copies.
General comments
In general immune-stimulatory effects of rChIL-17B in vitro and NC8-ChIL-17B in vivo can be supported by various experimental data but there are some drawbacks in animal experiment. - Although the forty 3-day-old chicks were randomly grouped into 4 groups their initial body weights were unignorably different especially in case of IBV group (Table 2). Body weights (BW) of newly hatched chicks is a barometer of health during embryos and chicks. We usually weigh all the chicks and divide into groups with similar average BW and standard deviation.
- The authors used acute infection model by testing viral RNA on 10 day-post-challenge, and they could not find significant difference between NC8+IBV and NC8-ChIL-17B+IBV groups (both groups showed significant viral RNA reduction). IBV persist several weeks in cecal tonsils (Hong et al., Viruses 2018, 10, 652 etc.), so chronic/persistent infection model testing viral RNA in cecal tonsils at 4 week-post-challenge may have been interesting.
Specific comments
- lactobacillus plantarum to Lactobacillus plantarum or L. plantarum in the manuscript.
- No need to repeat Lactobacillus plantarum when describing NC8-ChIL17B and NC8 in the manuscript.
- Please describe the specificity of primer set (only to M41) used for virus QRT-PCR. Did you verify the specificity of primer set by using M41 and H120 RNAs? To support the QRT-PCR result of trachea in PBS-IBV group.
- Need to correct typos and minor grammatical errors in the manuscript. Followings are examples.
- Line 17: ‘protein’ to ‘gene’
- Signal peptide sequence SPUsp45 is known to facilitate secretion of fusion protein out of the bacterial cells. rChIL-17B is secreted from NC8-ChIL-17B or only trapped in peptidoglycan layer?
- Did you use live or fixed (with what organic solvent?) NC8-ChIL-17B for whole cell ELISA?
- Line 213: outer-membrane? L. platarum is a Gram+ bacteria. Cytoplasmic membrane?
- Line 221: rChIL-17B; Line 227: rNC8-ChILIL17B to rChIL-17B. Please clarify usage of terms for consistency, recombinant ChIL-17B (rChIL-17B), NC8 strain expressing rCHIL-17B gene (NC8-ChIL-17B).
- Line 299: ‘cecum tonsils’ to ‘cecal tonsils’ and others in the manuscript
- Line 313: Spleen? maybe kidney?
- Line 354: ‘though’ to ‘through?’ line 359: on the other hand
- Supplement
Line 12: ‘lactobacillus plantarum NC8 protein’ to ‘NC8 protein’
Line 20: Please add concentration of each primer
Line 29-30, 32: ‘rChIL-17B stimulated’ to ‘rChIL-17B-stimulated’
Author Response
Response to Reviewer 1 Comments
Point 1: In general immune-stimulatory effects of rChIL-17B in vitro and NC8-ChIL-17B in vivo can be supported by various experimental data but there are some drawbacks in animal experiment. - Although the forty 3-day-old chicks were randomly grouped into 4 groups their initial body weights were unignorably different especially in case of IBV group (Table 2). Body weights (BW) of newly hatched chicks is a barometer of health during embryos and chicks. We usually weigh all the chicks and divide into groups with similar average BW and standard deviation.
Response 1: Thank you very much for your professional advice. We had ignored the uniformity of initial weight and only focused on the effect of sex on weight gain of chicken, because of the limit SPF chicks due to low hatchability (only 78%) of the SPF chicken embryos which was transported by train from Beijing to Sichuan for more than 28 h in China. Thus, to avoid the influence of initial weight on statistical results, we have redone statistics by unpaired t-test. The results showed that, the weight gain of chickens in the NC8-ChIL17B+IBV group was absolutely higher than chickens in NC8-P+IBV group at the end of the 28 days vaccination period by using unpaired t-test (p<0.0001), and the initial weights of the two groups had no significant different. The initial weight and end weight in PBS group and PBS+IBV group had no significant different In fact, we also have set the NC8-ChIL17B strain group and NC8-P strain group without IBV vaccine in order to evaluate the effect of the unique recombinant bacteria without along with vaccines on the growth performance and immune immunoenhancement effect of chicken ,But the results did not showing in this article. Then we showed the results as bellow table R1. The initial weights of NC8-ChIL17B, NC8-P and PBS groups have not significant difference, but at the end of 28 days vaccination period, the weights gain of chicken in NC8-ChIL17B group were very significantly different compared with the NC8-P and PBS groups (p<0.0001). these results indicated that the NC8-ChIL17B can promote the growth performance of treated chickens. And the potential mechanism was discussion in line 2335-2342.
Table R1. Chicken weights during the 28 days experimental period
|
Group |
Initial weight (g) |
End weight (g) |
Net gain (g) |
|
PBS |
62.0 ± 0.76 |
360.0 ± 5.79 |
298.0 ± 4.63C |
|
NC8-P |
62.00 ± 0.76 |
370.0 ± 4.70 |
308.0 ± 3.89B |
|
NC8-ChIL17B |
63.00 ± 0.26 |
405.0 ± 5.37 |
342.0 ± 4.26A |
Tukey's multiple comparisons test was used to statistics, n=10 in each group. The weight gains are shown as mean±SEM. The capital letters indicate the significant difference at p<0.0001.
Point 2: The authors used acute infection model by testing viral RNA on 10 day-post-challenge, and they could not find significant difference between NC8+IBV and NC8-ChIL-17B+IBV groups (both groups showed significant viral RNA reduction). IBV persist several weeks in cecal tonsils (Hong et al., Viruses 2018, 10, 652 etc.), so chronic/persistent infection model testing viral RNA in cecal tonsils at 4 week-post-challenge may have been interesting
Response 2: The reason for no significant difference between NC8+IBV and NC8-ChIL17B groups is that we used Tukey's multiple comparisons test for statistics. In the revised version manuscript, we took the multiple t-test for statistics. the new results are as follows: The mean IBV M41 copies in IBV immunized groups IBV, NC8-P+IBV and NC8-ChIL17B +IBV were very significantly reduced compared with the PBS group (p<0.01). The mean IBV M41 copies in trachea, lung, spleen, kidney and bursa from the group NC8-ChIL17B+IBV were very significantly or significantly lower compared with the control group NC8-P+IBV, (p<0.01 in lung, p<0.05 in trachea, spleen, kidney and bursa). The results indicated that there has significant difference between NC8-P+IBV and NC8-ChIL17B. In fact, on the 3rd, 6th and 9th dpc, we had collected cloacal swabs for monitoring the virus shedding by ordinarily PCR. On the 3rd, 6th dpc, it can be amplified M41 specific band but on the 9th dpc, we cannot detect specific band except the PBS group. And in previous study, the IBV RNA level in swabs on the 10th dpc dropped off than on the 8th dpc (Xuan Wu et al., Viruses, 2019, 11, 254). So, we considered to euthanize all the chickens and analyze the M41 copies in the tissues on the 10th dpc.
Point 3: - lactobacillus plantarum to Lactobacillus plantarum or L. plantarum in the manuscript.
Response 3: It was corrected in the newly revised manuscript. Line 3, 34, 78, 82, 88, 90, 391,393, 396, 400, 402, 424, 498, 506.
Point 4: - No need to repeat Lactobacillus plantarum when describing NC8-ChIL17B and NC8 in the manuscript.
Response 4: It has been revised in the revised manuscript. Line 21, 23, 24, 25, 29, 30, 79, 83, 86, 106, 107, 109, 125, 126, 129, 148, 151, 155, 157, 165, 166, 167, 173, 183, 250, 251, 252, 253, 256, 257, 266, 272, 273, 278, 281, 282, 283, 284, 294, 295, 296, 298, 299, 310, 311, 313, 315, 325, 325, 326, 327, 339, 340, 341, 343, 346, 359, 362, 364, 365, 401, 405, 423, 424, 432, 438, 441, 445, 453, 454, 456, 466, 467, 484, 488, 491, 492, 500, 502, 507…
Point 5: - Please describe the specificity of primer set (only to M41) used for virus QRT-PCR. Did you verify the specificity of primer set by using M41 and H120 RNAs? To support the QRT-PCR result of trachea in PBS-IBV group.
Response 5: The exact primers that can differentiate the H120 vaccines strain with violent M41 were shown in line 238-239. The previous manuscript was made a wrong copy from the primers pool. The primers that was used to detect the copies of M41 in tissues from experimental chicken were shown as follows: PIBV-F 5’- TCTGAGAAATCAGTTGAGGGT-3’; PIBV-R 5’-ACTCATCAACCTCTTCTGCTG-3’. The BLAST result of M41 complete genome and H120 complete genome is shown as Figure R1 (only part of variant section). The red boxes indicate primers position.
Figure R1. the result of BLAST
Point 6: - Signal peptide sequence SPUsp45 is known to facilitate secretion of fusion protein out of the bacterial cells. rChIL-17B is secreted from NC8-ChIL-17B or only trapped in peptidoglycan layer?
Response 6: 2.2 and Fig.1 show the construction strategy of rChIL-17B gene and pMG36e-ChIL17B plasmid. the signal peptide sequence SPUsp45 (MKKKIISAILMSTVILSAAAPLSGVYA) was linked to the ChIL-17B gene and we have predicted the amino acid sequence of the rChIL-17B gene by SignalP-4.1 Server online, as shown in Figure R3 as follows. The Cleavage site is between pos. 27 and 28 (VYA-GS), consistent with previous forecasts. And the probability was 0.8451. Moreover, we have predicted the amino acid sequence of the rChIL-17B gene by TMHMM2.0 online and found the signal peptide sequence SPUsp45 is just at the transmembrane region and the ChIL-17B is at the outside of the NC8 cell. As shown in Figure S1(a, b), the rChIL-17B protein the rChIL-17B is only anchored on the cell surface of NC8. Cell surface protein were extracted by SDS and 1% mercaptoethanol.
Figure R3. The predicted results of rChIL-17B. the cleavage site and cleavage probability were predicted by SignalP-4.1 Server online; the transmembrane region was predicted by TMHMM2.0 online
Point 7: - Did you use live or fixed (with what organic solvent?) NC8-ChIL-17B for whole cell ELISA?
Response 7: The NC8-ChIL-17B cell was resuspended and fixed with 4% paraformaldehyde (Biosharp life science, Hefei, China) for 30min in the whole cell ELISA assay, as shown in supplementary materials, line 25.
Point 8: - Line 213: outer-membrane? L. platarum is a Gram+ bacteria. Cytoplasmic membrane?
Response 8: The rChIL-17B protein was anchored to the NC8-ChIL-17B cell surface. It was corrected in line 125.
Point 9: - Line 221: rChIL-17B; Line 227: rNC8-ChILIL17B to rChIL-17B. Please clarify usage of terms for consistency, recombinant ChIL-17B (rChIL-17B), NC8 strain expressing rCHIL-17B gene (NC8-ChIL-17B).
Response 9: In the revised manuscript, the terms have been clarified consistency.
Point 10: - Line 313: Spleen? maybe kidney?
Response 10: It is kidney. But in the revised manuscript the sentence has been deleted.
Supplement comment
Point 11: Line 20: Please add concentration of each primer
Response 11: The concentration of each primer was 10 μmol /L, see line 220 in the article.
Point 12: Line 29-30, 32: ‘rChIL-17B stimulated’ to ‘rChIL-17B-stimulated’
Response 12: In the new version of supplement, see line 61, 68.

Reviewer 2 Report
The manuscript entitled “Construction and immunoadjuvant activities of the oral interleukin-17B expressed by lactobacillus plantarum NC8 on IBV vaccination of chicken” by Guo et al. reported that lactobacillus planetarium surface expressed chicken IL-17B can be used as oral vaccine adjuvant to improve the live attenuated IBV vaccine efficacy. It’s an interesting study, however, the design, organization and writing of this manuscript make it unacceptable to be published in this current form. Below are some points that should be revised.
- Many typos and mistakes in the manuscript make it difficult to understand. The misspelling of kDa in Line 20, CD8 in line 28, 270, 276 and many other places suggested that authors did not check this manuscript carefully before submission.
- In line 87-88, “The rChIL-17B was design as Fig1”. However, no such information was given in Fig 1.
- Also in Fig 1b, please show the protein molecular weight.
- The name of groups should be consistent throughout the manuscript. For example, NC8-ChIL-17B in Line 227 was named as NC8-ChIL17B in the Fig 2. And in Table 2, Fig3 and Fig4, the group “IBV” became “PBS+IBV” in Fig 5 6 7 and 8. They are confusing.
- The authors did not mention the multiple comparison tests they used in the statistical analysis.
- In Line 138, the authors said that 1e9 CFU of NC8-ChIL17B was given to each chicken, what’s the exact mass of IL-17B in such dosage?
- Which gene does the primer pair targeting to in Line 205-206? Is it special enough to differentiate the H120 vaccine strain with M41 challenge strain?
- In Section 2.5.2, the authors only mentioned about the kit to determine the total sIgA. While in the Section 3.4.4, they showed total IgA and IBV specific IgA results. The method to determine IBV specific IgA should be contained in the material and methods part.
- Why did the authors select thymus as an analysis target in Section 3.4.5? Generally, thymus is a primary lymphoid organ and it’s not a good target to analyze vaccine efficacy. Maybe, other mucosa associated chicken secondary lymphoid tissues such as Harderain gland and pulmonary lymphocytes are more suitable for this analysis.
- In line 148-149, spleen, tracheal lymph node, bursa and small intestines were collected to analyze the immune related genes. However, in Line 182, tissues used to analyze the immune related genes change to thymus and cecum tonsil. Where are the other tissues? And in Line 154-155, lung, liver, spleen, kidney bursa and small intestines were collected for M41 existence detection. However, in Fig 8, the tissues are different with those mentioned before.
- Also, it’s necessary to present the virus detection from throat and cloacal swabs, since reduction of virus shedding is important to evaluate an IBV vaccine.
- And in Line 313, “IBV M41 copies in spleen were low”, it should be kidney not the spleen according to their figure.
Reviewer 3 Report
The study entitled "Construction and immunoadjuvant activities of the 3 oral interleukin-17B expressed by lactobacillus plantarum NC8 on IBV vaccination of chicken" is very interesting, complete and preludes to the implementation of and adjuvant for the IBV vaccine in ckicken to increase the efficacy of the vaccine and avoid adverse effects. However the manuscript needs to be thoroughly improved in language and presentation according to the following indications.
Whole manuscript: for this manuscript it is extremely necessary to correct the English style by aid of a proficient mother tongue scientific proofreader since it contains many grammatical mistakes (this mostly from Materials and Methods on and I do not give examples because mistakes are too many), check spaces, character style and font size, italicize "in vitro", lactobacillus plantarum with first letter of lactobacillus capital and spelled out only at its first occurrence in the manuscript (always italicized), detail better Materials and Methods (see examples below), insert a space between the number and μl or μg or other measure units,
Examples of specific remarks
Line 27: 47 not "forty 7"
Line 71: can induce
Line 77: we cloned chicken IL-17B into...
Line 80: has SPF to be spelled out here?
Line 83: "vector" commonly means a plasmid used to clone genes, better to say "the bacterial host used to express and deliver the recombinant..."
Line 85: add MRS supplier
Line 87: the oritein of interest
Line 95: give a reference for the pMG36e plasmid
Line 96: add supplier details for E. Coli DH5α
Line 97: what is SOC (spell out)? Composition?
Line 98: add electrotransformation parameters and equipment
Line 99: cultured on MRS agar added with10 μg/mL erm; what do you mean with "positive strain"? Please, explain
Line 102: preliminary results...
Line 106: culture volume?
Line 110: pellet amounts? Lysozyme amount?
Lines 114-116: detail electrophoretic conditions and Western blot procedure
Line 124: you did not specify above that you purified the protein
Line 126: procedure of the cytotoxicity assay?
Line 131: spell out TCID50
Lines 136, 214-215, 217 and whenever it occurs: not "empty vector" but "wild type" L. plantarum NC8
Line 192: describe the negative control and explain what did you use as positive controls for the immunity-related genes
Loine 180: add BioRad manufacturer or distributor details
Lines 182-183: how did you homogenate tissue? How was Trizol extraction carried out?
Line 190: sterile ddH2O
Line 202: or always qPCR or always QPCR, regardles if RT or not
Lines 205-206: PCR parameters?
Lines 213-214: fact of OD630 not well written
Figure 1: the ordinate axis must be entitled just as OD630; isn't the protein determined by ELISA too high in the wild type?
Line 253: re-list the cytokines here
Line 266: the abbrevietion FCM must appear first in brackets nearby its spell out in materials and methods
Line 316: standard curve (the figure legend is not in the correct format)
Line 318: gram?
Line 321: not all L. plantarum strains have no plasmids
Line 326: italicize Eimeria tremella
Line 366: use the abbreviation of days post...
Discussion: in general, do not reiterate results
Conclusions: what are the perspectives for your study?
Supplementary material:
Please, check spaces and special characters
Reviewer 4 Report
Comments to the author:
In this study, Guo et al. explored the immunoadjuvant potential of NC8-ChIL17B on IBV vaccination of chicken. The recombinant NC8-ChIL17B could efficiently enhance the humoral and cellular immune responses to IBV vaccine by elevating immune gene expression and induced proinflammatory cytokine induction. Some comments below.
- Figure 1 is to describe the validation of expression of CHIL-17B. It could be put into supporting information.
- Did not see the data of the RNA level change of JAK2, TYK2, STAT1, STAT3 related to JAK/STAT signal pathway? Any figure missing?
- It’s difficult to understand in Figure 2a, why treating more rNC8-ChIL17B protein led to weaker effect compared with 50ng group in infected HD11 cells?
- Is there any rational explanation why for some immune-related genes, the expression level was upregulated in IBV and NC8+IBV group compared with PBS control, but some are not? In Figure 7, takes IL22 for example. These is huge difference in cecum tonsil or thymus.
- The numerous minor mistakes during the writing should be carefully checked. The current format could not be accepted.
i.e. Page 1 Line 18 and 20: Lactobacillus plantarum should be italicized throughout the manuscript. Same applied to Line 21 “in vitro” etc.
Page 1 Line 19: Uniform the font in Abstract.
Missing literature information (page number etc.) on Reference section like for No. 49 Reference.
Page 5 Line 213 & 216, uniform the writing style whether the authors want to capitalize it or not “Figure1a” “figure1b”
Page 6 Line 239 p < 0.001 should be denoted as *** .
Figure 2a “protein” should be protein
Page 8 Line 270: CD4+/C8+ should be CD4+/CD8+
Incorrect y-axis title in Figure 4
Incorrect y-axis title in Figure 5A
Round 2
Reviewer 2 Report
I can't find the "figure R1" as indicated in authors' response. This manuscript has not improved in terms of the scientific validity. The responses to comments are sloppy. Questions have not been appropriately addressed. Extensive editing of English language and style are required.
Author Response
1. The manuscript was checked by MDPI English editing service and I have confirmed the English-edited version carefully.
2. details of the revisions
Line 2: the title was added “the” before “construction” and “IBV”; “chicken” to “chickens”;
last paragraph of “Introduction” was added “Probiotic Lactobacillus, which can tolerate gastric acid and bile salts and stay in the intestines for several days, are considered ideal carriers for mucosal administration. L. plantarum NC8 is a nonpathogenic, non-sporulating, non-plasmid bacterium that has been reported to be an effective delivery system for HA2 of the H9N2 influenza virus [38], the H1N1 M2e antigen [39], and the HN of Newcastle disease virus (NDV) [40] in chickens. Moreover, oral administration is kinder to chickens, and saves time and labor under intensive conditions.”
The first paragraph of “Discussion” was deleted due to it was not sticking to our observation.
Line 129: in Figure 1. added the gene or other elements names in the figure legend.
Figure 3, 4, 5: “day post vaccination” to “dpv”
Line 307: in Figure 3, the S/P was spelled out.
Reviewer 3 Report
Reviewer report 2
The manuscript now is more correct in scientific content but language still needs improvement. Examples are below and the next ones are omitted because I was tired of writing down the language imprecisions or unclear information every two or three lines.
Spaces between words and punctuation are still not correct
Line 60: response
Line 74: can induce
Line 75: to defend
Line 79: we cloned chicken IL-17B
Lines 85-90: correct line spacing
Line 86: say the host for what
Line 105: cultured
Line 109: naming what?
Line 110-111: this sentence is not complete
Line 112: I do not think the “identified primers” is a correct definition
Line 127: anchored
Line 130: centrifuged
Line 133: was subjected
Line 136: what does “flowing the instrument” mean?
Line 139-141: the verb is missing
Line 147: CO2 with subscript
Line 148: until they covered
Lines 149-150: and resuspended (you already have the verb above)
Line 152: invert subject and verb
Line 156: followed by?
Correcting the English language means that the manuscript must be corrected by a proficient English proofreader, preferably mother tongue and preferably expert of scientific English, it does not mean that an author alone has to try to make the text correct if he/she is not yet perfect in English grammar; it is impossible to succeed also because the manuscript is complex. Just ask your institution to pay for English revision as all of us authors have to do. This is the surest way to really improve the paper.
I say it already now: I appreciate very much this study but if the manuscript is returned to me again without acceptable English I will recommend rejection next time.
More about content
The answers given to my previous objections must be all reported in the text, except the function of the SOC medium. The comment 154 that you did not understand maybe meant just that the correct wording “protein” was required.
Line 124: you have to explain gene or other elements names also in the figure legend. The reader has to be able to interpret the figure without referring to the main text.
Line 290: please, spell out S/P, I don’t think everyone e knows what it is. It is more appropriate to insert this in short in the materials and methods.
Figures 3, 4 and 5: you use dpv throughout the whole text, but in the figures it is spelled out. This is not coherent
Line 360: spell out dpc at first occurrence
Line 368: standard
DISCUSSION: this section should be shorter. You have to explain your results with those of others sticking to your observations. This means the you have to re-elaborate the section by starting each paragraph with a brief resume of your results, though not reiterating them, and commenting each result by reference to previous reports and not considering observations and literature that are not strictly related to your observations.
Line 373: here and elsewhere “diseases”
Lines 380-392: this is more appropriate for the introduction where you can merge these concepts with what already presented without reiterating what you have already said. 391-392: this applies to not NC8 but to L. plantarum in general
Author Response
The manuscript was checked by MDPI English editing service and I have confirmed the English-edited version carefully. And The details of revisions and responses to reviewers' comments are showed below:
details of the revisions
Line 2: the title was added “the” before “construction” and “IBV”; “chicken” to “chickens”;
last paragraph of “Introduction” was added “Probiotic Lactobacillus, which can tolerate gastric acid and bile salts and stay in the intestines for several days, are considered ideal carriers for mucosal administration. L. plantarum NC8 is a nonpathogenic, non-sporulating, non-plasmid bacterium that has been reported to be an effective delivery system for HA2 of the H9N2 influenza virus [38], the H1N1 M2e antigen [39], and the HN of Newcastle disease virus (NDV) [40] in chickens. Moreover, oral administration is kinder to chickens, and saves time and labor under intensive conditions.”
The first paragraph of “Discussion” was deleted due to it was not sticking to our observation.
Line 129: in Figure 1. added the gene or other elements names in the figure legend.
Figure 3, 4, 5: “day post vaccination” to “dpv”
Line 307: in Figure 3, the S/P was spelled out.
Reviewer 4 Report
The authors have addressed my concerns.
Author Response
The manuscript was checked by MDPI English editing service and I have confirmed the English-edited version carefully.